# Development of Plant-Based Vaccines for Prevention of Avian Influenza and Newcastle Disease in Poultry

**DOI:** 10.3390/vaccines10030478

**Published:** 2022-03-19

**Authors:** Ika Nurzijah, Ola A. Elbohy, Kostya Kanyuka, Janet M. Daly, Stephen Dunham

**Affiliations:** 1School of Veterinary Medicine and Science, University of Nottingham, Sutton Bonington Campus, Loughborough LE12 5RD, UK; ika.nurzijah@nottingham.ac.uk (I.N.); ola.elbohy@nottingham.ac.uk (O.A.E.); janet.daly@nottingham.ac.uk (J.M.D.); 2Biointeractions and Crop Protection, Rothamsted Research, Harpenden AL5 2JQ, UK; kostya.kanyuka@niab.com; 3Faculty of Pharmacy, Universitas Muhammadiyah Purwokerto, Purwokerto 53182, Indonesia; 4Department of Virology, Faculty of Veterinary Medicine, Mansoura University, Mansoura 35516, Egypt; 5National Institute of Agricultural Botany (NIAB), 93 Lawrence Weaver Road, Cambridge CB3 0LE, UK

**Keywords:** plant-based vaccines, avian influenza virus, Newcastle disease virus, haemagglutinin protein, *Agrobacterium tumefaciens*, *Nicotiana benthamiana*, transient expression, virus-like particles

## Abstract

Viral diseases, including avian influenza (AI) and Newcastle disease (ND), are an important cause of morbidity and mortality in poultry, resulting in significant economic losses. Despite the availability of commercial vaccines for the major viral diseases of poultry, these diseases continue to pose a significant risk to global food security. There are multiple factors for this: vaccine costs may be prohibitive, cold chain storage for attenuated live-virus vaccines may not be achievable, and commercial vaccines may protect poorly against local emerging strains. The development of transient gene expression systems in plants provides a versatile and robust tool to generate a high yield of recombinant proteins with superior speed while managing to achieve cost-efficient production. Plant-derived vaccines offer good stability and safety these include both subunit and virus-like particle (VLP) vaccines. VLPs offer potential benefits compared to currently available traditional vaccines, including significant reductions in virus shedding and the ability to differentiate between infected and vaccinated birds (DIVA). This review discusses the current state of plant-based vaccines for prevention of the AI and ND in poultry, challenges in their development, and potential for expanding their use in low- and middle-income countries.

## 1. Introduction

Poultry are a major source of animal protein, particularly chicken. The worldwide chicken population is over 20 billion birds and production systems range from intensive units, containing over 100,000 birds, to small backyard flocks. Poultry provide not only a valuable supply of dietary protein but also an important source of income in rural areas of developing countries. Viral diseases continue to threaten poultry production and cause significant economic loss through mortality and reduced growth. Avian influenza (AI) and Newcastle disease (ND) are the most prevalent viral infections in poultry. Given their economic and societal impact, both ND and some forms of AI are notifiable to the World Organisation of Animal Health (OIE). Control of AI and ND relies heavily on vaccination, and intensive systems underpin this with high levels of biosecurity. Despite the availability of AI and ND vaccines, outbreaks of AI and ND will likely persist due to issues with commercial vaccines. Commercial AI and ND vaccines are expensive, require cold chain storage, and often poorly protect against local emerging strains, which limits their benefits for low-resource markets. Therefore, there is an urgent need for vaccines that can be produced at relatively low cost, are stable and can be readily adapted to local virus variants. Plant-based vaccines offer such advantages and are suitable for use in developing economies. This review addresses the current state of plant-based vaccines for the prevention of AI and ND, challenges in their development, and potential for expanding their use in low- and middle-income countries. In particular, we highlight the development of plant-based virus-like particle (VLP) vaccines against AI and ND.

## 2. Avian Influenza Virus (AIV)

Influenza viruses belong to the family *Orthomyxoviridae*, which is comprised of seven genera [1]. Avian influenza is caused by the species *Influenza A virus* (IAV) in the genus *Alphainfluenzavirus*. Virus particles are spherical, approximately 100 nm in diameter, or filamentous, about 300 nm in length [2]. The virus particle is covered with glycoprotein spikes of haemagglutinin (HA), which represents nearly 80% of the total surface proteins, and neuraminidase (NA), which represents 17% of the total surface proteins (Figure 1). Matrix protein 2 (M2) is a minor surface protein, with around 16 to 20 molecules per virus particle. The host-cell-derived lipid membrane covers a matrix of M1 protein, which surrounds the virus particle core. Within the M1 matrix are the nuclear export protein (NEP) and the ribonucleoprotein (RNP) complex, which is composed of eight viral negative-sense RNA segments covered with nucleoprotein (NP) and the RNA-dependent RNA polymerase (RdRp), comprised of two basic and one acidic polymerase subunit (PB1, PB2, and PA).

Avian influenza viruses (AIV) are classified into 16 HA and 9 NA subtypes, which occur in many different combinations (e.g., H5N1). Due to the segmented nature of the RNA genome, reassortment can occur during co-infection with different AIV subtypes. Where this involves exchange of the HA and/or NA segments, it is referred to as “antigenic shift”. In addition, replication is error prone and the viral polymerase lacks proofreading activity, resulting in considerable genetic drift over time, which can lead to “antigenic drift” and the ability of emergent viruses to escape natural or vaccine-induced immune responses [3]. Avian influenza viruses can also be classified into two pathotypes. Low pathogenic avian influenza (LPAI) viruses are the most common but, nonetheless, can have a significant economic impact. During the late 1990s, poultry-adapted H9N2 became endemic in several different countries in the Middle East, Asia, Africa, and Europe. In chickens, LPAI H9N2 infections can produce mild to intense respiratory disease signs, significant economic loss due to reduced egg production, elevated rates of morbidity, and up to 20% mortality [4]. Occasionally, mutation in the HA of H5 and H7 LPAI strains results in the acquisition of a polybasic cleavage site, giving rise to highly pathogenic avian influenza (HPAI) viruses. These produce intense, generalised disease in chickens, turkeys, and other gallinaceous poultry; mortality can reach 100% in a few days. In acute cases, lesions include cyanosis and oedema of the head, comb and wattle; oedema and inflamed shanks and feet due to subcutaneous haemorrhages; petechial haemorrhages on visceral organs and in muscles; and bloody oral and nasal discharges [5]. On the other hand, in peracute cases, death may occur in the absence of clinical signs.

## 3. Newcastle Disease Virus

The causative agent of ND was initially called Newcastle disease virus (NDV). After classification as a member of the family *Paramyxoviridae*, it was renamed *Avian paramyxovirus-1* (genus *Avulavirus*). However, the ICTV recently reclassified paramyxoviruses based on phylogenetic distances between the complete large (L) protein amino acid sequences. Therefore, the official nomenclature of the species that causes ND is *Avian orthoavulavirus 1* (AOaV-1), genus *Orthoavulavirus*, subfamily *Avulavirinae* [1]. Nonetheless, NDV is still in common usage.

Based on disease severity in chickens, NDV strains are further classified into four pathotypes: (i) asymptomatic enteric (considered as clinically non-problematic); (ii) lentogenic (causing subclinical to mild respiratory infections in younger birds); (iii) mesogenic (causing respiratory infection with low mortality); and (iv) velogenic (causing high morbidity and up to 100% mortality). Velogenic viruses can be further divided into two categories: viscerotropic velogenic viruses cause acute lethal infection and occasional haemorrhagic lesions in the intestines, and neurotropic velogenic viruses cause neurological and respiratory disorders [6,7]. NDV can be transmitted to healthy birds through oropharyngeal secretions and faecal matter. Susceptible birds can be infected by inhaling contaminated dust or aerosolised virus or by the ingestion of virus shed in bird droppings.

NDV is an enveloped virus with a non-segmented negative-sense RNA genome. The genome of NDV encodes six structural proteins: nucleoprotein (NP), phosphoprotein (P), matrix protein (M), fusion protein (F), haemagglutinin-neuraminidase (HN), and large polymerase (L) (Figure 2). The HN, F, and M proteins are tightly linked to the viral envelope. Anchored to and protruding from the viral envelope are HN and F glycoproteins. HN and F mediate viral entry into the host cell and virus particle release. Furthermore, upon infection, neutralising antibodies are directed against both the HN and F proteins [8].

The non-glycosylated M protein is peripherally attached to the inner surface of the viral envelope and involved in the morphogenesis and budding of NDV. NP is the most abundant protein in NDV particles. NP encapsidates the RNA genome to protect it from host nucleases. Each NP subunit is predicted to be associated with six nucleotides of RNA. The combination of NP-RNA is termed “nucleocapsid core”. Two additional viral proteins, P and L, are bound to the core forming a herringbone-like ribonucleoprotein (RNP) complex. The RNP complex can be visualised with electron microscopy using negative staining [10]. The RNP complex is associated with transcription and replication processes, which determine NDV virulence [11].

The nucleocapsid RNA serves as a template for transcription and replication by the viral RdRp, which consists of L and P proteins [8]. Apart from these structural proteins, the accessory proteins, V and W, are generated during *P* gene transcription by means of RNA editing in the virus-infected cells [12]. V protein has been suggested to direct host-immune evasion upon NDV infection, whereas the function of the W protein remains elusive [13,14,15].

## 4. Overview of Vaccines for AI and ND Immunisation

The different types of vaccine licensed or under development for AI and ND may be classified into six groups: (1) inactivated, (2) live-attenuated, (3) subunit, (4) vector-based, (5) DNA, and (6) VLP (Table 1).

Inactivated vaccines have a long history of use for the control of AI; similarly, ND vaccines have been used since the 1940s [16]. A number of inactivated AI vaccines are commercially available. These include monovalent inactivated vaccines comprising either H5 or H7 strains, bivalent vaccines with H5 and H7 strains, and both monovalent and bivalent vaccines with homologous or heterologous NA [17]. Both live-attenuated and inactivated vaccines, developed from non-pathogenic and lentogenic NDV isolates (e.g., LaSota and Hitchner B1), are the most commonly administered for the control of NDV [7,18]. However, there is evidence that LaSota-based vaccines are no longer effective against newly evolved NDV strains [7,19]. Protection afforded by inactivated vaccines depends on the quantity of antigen in each dose and how well matched the vaccine is to circulating viruses. Inactivated vaccines also require the use of an appropriate adjuvant. Live-attenuated virus vaccines have the potential for reversion to virulence and recombination with field virus. Consequently, live-attenuated AI vaccines against any subtype are generally not recommended for use in poultry. However, the use of both live-attenuated and inactivated vaccines does not allow for easy differentiation of infected from vaccinated animals (DIVA), which can make diagnosis and control more difficult [20].

The most rationally designed vaccines to meet the full criteria for an excellent NDV vaccine are the recombinant genotype-matched live-attenuated vaccines. The vaccine candidates are generated by reverse genetic technology and attenuated by modification of the multibasic cleavage site to a monobasic site in the F protein. These vaccines can protect against circulating virulent strains in certain regions and significantly reduce the viral shedding. In addition, they are compatible with a DIVA strategy. However, genotype-matched live-attenuated vaccines are not widely used as these are often geographically specific and require cold chain storage [9].

Subunit vaccines are efficient at inducing humoral and cellular immune responses against specific viral proteins without the risk of handling live virus during vaccine production or reversion to virulence. However, subunit vaccines have a more restricted antigenic repertoire than attenuated viral vaccines; as such, it is important to ensure that the elicited immune responses provide robust protection against viral challenge. For example, an ND subunit vaccine that is derived from a single glycoprotein may be poorly immunogenic; the use of both NDV F and HN leads to a broader antibody response that provides greater protection against viral challenge [21]. In addition, subunit vaccines may be poorly immunogenic due to misfolding of protein or poor identification by the immune system.

Viral vector-based vaccines for NDV induce strong humoral and cellular immune responses. The common poultry viruses, such as *Fowlpox virus* (FPV), *Herpesvirus of turkeys* (HVT), and *Infectious bursal disease virus*, have been used as vectors for expressing and delivering NDV F and HN proteins in chickens [22]. Recombinant FPV (the main vector used in approved AI vaccines) and HVT have also been used for AI vaccines as well as recombinant NDV containing H5 or H7 AIV gene inserts. Vector-based vaccines can be delivered by aerosol spray or eye drops at the hatchery to minimises administration costs. The use of FPV as a vector is only limited to chickens, and they must be naïve to the fowlpox vector for immunisation to be efficient. Pre-existing maternally derived antibodies against the virus vector can inhibit the replication of the vaccines, limiting the immune response. This is essentially the major hurdle for application of vector-based vaccines [20,23].

**Table 1 vaccines-10-00478-t001:** Common types of vaccine and their respective advantages and disadvantages; modified from [24].

Vaccine	Advantages	Disadvantages
Inactivated vaccine 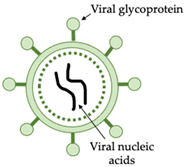	Safe Good humoral responses	Low immunogenicity Transient immunity Frequent and multiple doses may be needed Poor stimulation of cellular immunity Poor DIVA * compatibility Requires direct inoculation
Live-attenuated vaccine 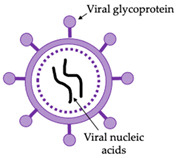	Contain important epitopes Single dose sufficient Enhanced cellular immunity	Safety hazard in immunocompromised birds Reversion to virulence Interference by maternal antibodies Poor DIVA * compatibility
Subunit vaccine 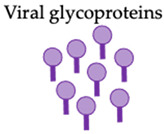	Safe Choice of most effective antigen	Low immunogenicity Transient immunity Poor stimulation of cellular immunity Requires direct inoculation
Vector-based vaccine 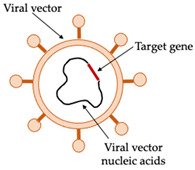	Safe Contain important epitopes Choice of most effective antigen Stimulation of humoral and cellular immunity Effective in heterologous prime-boost	Often multiple doses required Interference by vector-specific immunity
DNA vaccine 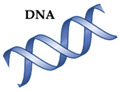	Contain important epitopes Choice of most effective antigen Stimulation of cellular immunity Efficient in heterologous prime boost	Usually, poor protection without adjuvant Humoral responses can be weak
Virus-like particles (VLPs) 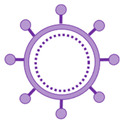	Safe Contain important epitopes Stimulation of humoral and cellular immunity	Complex production process

* DIVA, differentiation of infected from vaccinated animals.

DNA vaccines, in which gene segments encoding viral proteins are cloned into an expression plasmid, have been in development for AI and ND for 15 to 20 years. They offer the advantages of being safe (non-infectious) and effective at inducing cell-mediated immunity, but they are poor at inducing antibody responses, typically require multiple doses, and are therefore unsuitable for mass immunisation of poultry [9]. Non-replicating, haemagglutinin-based H5 RNA particle, H5 expressed baculovirus, and H5 DNA vaccines have been licensed for use in chickens since 2015; however, their usage has been limited [5].

VLPs form structures resembling that of the target virus particles but are unable to replicate because they lack nucleic acid. The particulate nature of VLPs enable their uptake by dendritic cells (DCs) for processing and presentation by major histocompatibility complex (MHC) class II. Their self-adjuvanting effect stimulates DC maturation and migration, leading to innate immune response stimulation [25,26]. Many VLPs that maintain virus receptor binding regions are capable of targeting and penetrating cells via their normal receptor. VLPs can therefore be processed via the MHC class I pathway, leading to the activation of CD8+ T-cells; such a cellular immune response is often critical to the control of infections by intracellular pathogens such as viruses. The ability of VLPs to target DCs is a significant feature of VLP vaccines that is crucial for the stimulation of innate and adaptive immune responses. Due to potent activation of dendritic cells, VLPs are less susceptible to immune tolerance [27].

The functional NDV VLP requires the assembly of viral envelope proteins consisting of F, HN, and M proteins, which complicate the production steps without the appropriate expression system [28,29]. On the contrary, influenza HA is able to self-assemble into VLPs under suitable conditions. The technical challenges for the generation of VLPs from enveloped viruses (enveloped VLPs/eVLPs) such as AIV and NDV remain a major hurdle to the success of this technology going forward. The stability of eVLPs requires thorough considerations of the appropriate temperature, shear force, and processes that are used in purification. Improper purification may lead to degradation of VLPs and a consequent reduction in immunogenicity. Most often, this leads to more complex downstream processing than conventional vaccines [30]. Alternative approaches for the assembly of VLPs are also being investigated, including the development of “hybrid” VLPs, which use the core of one virus as a scaffold for the surface glycoproteins of another virus. Hepatitis B cores (HBc) are one example of this, where the HBc structure has been used as a VLP core to present surface antigens from heterologous viruses including HIV, foot and mouth disease virus, and influenza A (reviewed in [31]). Such hybrid VLPs may offer a universal platform for VLP production and have the advantage of producing non-enveloped particles, which are inherently more stable than enveloped VLPs [32].

However, the technology for eVLPs production has shown significant advances. Recently, the Canadian biotechnology company Medicago reported that Health Canada approved the new drug submission (NDS) for a plant-based seasonal human influenza recombinant vaccine for scientific review. This is a quadrivalent virus-like particle (QVLP) vaccine including the HA of two influenza A virus subtypes and two lineages of influenza B virus transiently expressed using *Agrobacterium tumefaciens*-mediated transient transformation (also known as ‘agroinfiltration’). The QVLP vaccine elicited both homologous and heterologous antibody responses at higher doses, with the introduction of adjuvant having little or no effect. In the first phase III efficacy study, the candidate vaccine also reliably elicited both homologous and heterologous CD4+ T helper cells identified by generation of interferon-gamma (IFN-γ), interleukin-2 (IL-2) and/or tumour necrosis factor alpha (TNF-α) [33].

## 5. Pipeline for Production of Plant-Based Vaccines

The production of plant-based virus vaccines involves the transfer of the target virus gene into the plant for protein expression (Figure 3). The initial step is the choice of antigen from the virus. Viral glycoproteins are excellent antigens for veterinary vaccines as they are usually the main targets for virus-neutralising antibodies.

The target gene can be amplified via PCR-based methods, but usually, the gene is synthesised, which removes the need to isolate and handle highly pathogenic viruses. The design of the expression construct usually involves codon optimisation of the virus gene for expression in plants to increase the protein yield. Nevertheless, optimising an increasing percentage of codons does not necessarily lead to an improvement in protein espression levels above a certain level. Moreover, the extent of codon optimisation required may vary depending on the gene [34]. Undesired introns and RNA motifs can also be removed if the gene is synthesised. Flanking restriction sites should be added to ease the cloning process. There are some additional sequences that may be added at the 5′ untranslated region (UTR) such as a Kozak sequence to enhance initiation of translation of the gene [35] and a signal peptide to direct protein translocation. Typically, a peptide tag such as six histidine residues (6-His) is added at the carboxy terminal to enable affinity-based purification.

Protein expression can be achieved by transient or stable transformation of plants (reviewed in [36]). In stable transformation, the desired gene is permanently incorporated into the plant cell genome via nuclear or plastid integration to generate transgenic lines. Transformation is achieved using *A. tumefaciens* or, where this is not possible, biolistic or microparticle bombardment. The disadvantage of this system is that it is more time consuming, the protein yield is relatively low, and the random nature of entry into the plant nucleus can affect the transgene expression due to epigenetic regulation and silencing phenomena. Post-transcriptional gene silencing (PTGS) is a natural defence of the plant against pathogenic or heterologous RNA, and this phenomenon affects the transgene expression levels. Co-expression of silencing gene suppressors such as Cucumber mosaic virus (CMV) 2b or Tomato bushy stunt virus (TBSV) p19 with the transgene leads to improved expression levels in plants [37,38,39,40]. In contrast, transient transformation involves the production of the desired protein without integration into the plant cell genome. Transient expression systems may also use agroinfiltration for delivery of the vector containing the desired gene into plant cells for recombinant protein expression [41]. Agroinfiltration refers to the introduction of the *A. tumefaciens* containing the vector into the extracellular spaces of the plant leaf by a needleless syringe or under a vacuum (Figure 3). Transient expression is a simple method that allows the production of high yields of protein in as little as 3 to 7 days [42,43]. Transient expression also diminishes regulatory and public concerns regarding the use of genetically modified plants and transgene spread via pollen or seeds.

**Figure 3 vaccines-10-00478-f003:**
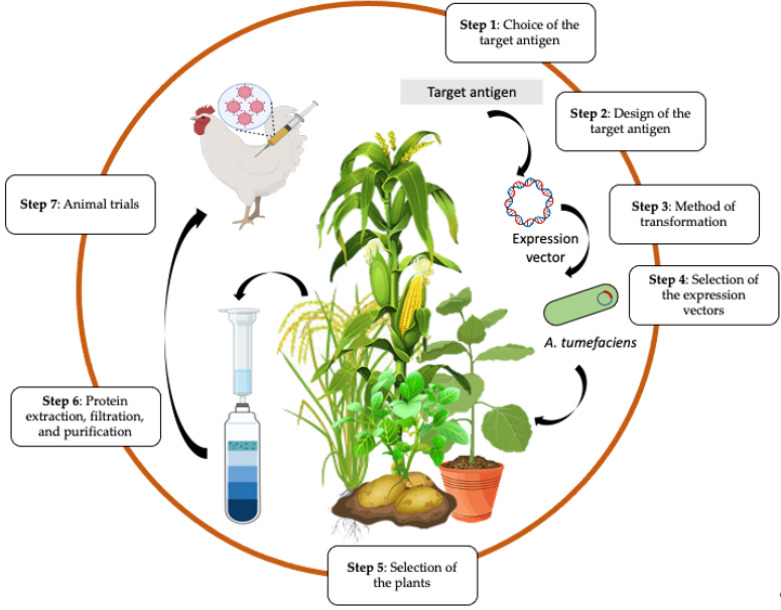
Summary of steps for production of plant-based vaccines. Created using some elements from Biorender.com (modified from Takeyama et al. [44]).

*A. tumefaciens*, which is a soil bacterium that is responsible for tumour formation in plants, has been modified for use in the delivery of desired genes without tumour formation [45] by disabling the tumour-controlling genes of the Ti plasmid. A variety of ‘disarmed’ *A. tumefaciens* strains with different marker genes have been developed that may have differing translational efficiencies in different plant species [46]. The addition of acetosyringone, a potent enhancer of *A. tumefaciens* virulence genes, in the infiltration buffer can improve the transformation efficiency and recombinant protein production levels [39]. The density of the bacterial culture is a crucial factor to consider. Low culture densities lead to low protein yields, while high densities can lead to host cell death [46].

Binary vectors are the standard of choice for the generation of transgenic plants. The term binary reflects the combination of T-DNA binary vector and *vir* helper plasmid that originate from the disarmed Ti plasmid of *A. tumefaciens*. A convenient feature of binary vectors is that they accommodate replication in multiple hosts (*E. coli* and *A. tumefaciens*). The T-DNA portion of binary vectors is flanked by left and right border sequences; this portion also consists of multiple cloning sites and plant selectable markers. The *vir* gene is located outside the T-DNA portion and is important to facilitate transfer and integration of T-DNA into the plant genome. Adjacent to the *vir* gene are bacterial selectable markers and a bacterial origin of replication (*ori*) [47].

Alternatively, plant virus vectors designated as first-generation virus expression vectors are also used for stable genetic transformation. This system is essentially an entire virus genome encoding the gene of interest, which may be fused to the viral coat protein (CP) to avoid vector instability. The first-generation expression vectors present several limitations, particularly the need of multiple promoters and size constraints [48].

The use of second-generation expression plasmids that can carry larger size genes have overcome problems of low protein production [41,49]. These “deconstructed vectors” contain the gene(s) of interest and the essential plant virus elements needed for replication [50]. Typically, for deconstructed vectors, the CP gene is removed from the viral backbone along with modification of the 3′ and 5′ UTRs as well as the addition of a suppressor of gene silencing [51]. Many plant viruses have been used as viral backbones for deconstructed vectors including *Tobacco mosaic virus* (TMV), *Turnip vein clearing virus* (TVCV), *Potato virus X* (PVX), *Cowpea mosaic virus* (CPMV), and the DNA genome geminivirus *Bean yellow dwarf virus* (BeYDV) (Table 2). The pEAQ-HT vector is based on CPMV and produces high levels of proteins through high translational efficiency without virus replication [52]. The pEAQ-HT vector was further modified to generate pHREAC, in which synthetic 5′ UTRs enhance expression. pHREAC is also designed with type IIS restriction enzymes, for easier cloning [43].

The expression method, protein yield, and cost are factors to consider when selecting plants to use for expression [59]. They should have a rapid growth rate and elevated protein production. Potato (*Solanum tuberosum* cv. Kennebec) and tobacco (*Nicotiana* spp.) are commonly used for production of recombinant proteins because they are amenable to genetic alteration. *Nicotiana benthamiana* is frequently used for recombinant protein expression due to its susceptibility to most of the plant viruses as a result of a defective form of host RdRp [60]. The interest in developing edible vaccines in low-and middle-income countries also motivates the use of other staple crops such as corn (*Zea mays*) and rice (*Oryza sativa*).

The downstream processing of recombinant protein(s) expressed in plants includes protein extraction, filtration, and purification to remove plant tissue contaminants [61]. Protein extraction begins with grinding of infiltrated leaves in the presence of an extraction buffer, which should be optimised with the appropriate pH, salinity, and supplemented with protease inhibitors to ensure protein stability and prevent proteolysis [62,63,64]. Leaves can be dried prior to protein extraction to reduce the volume of buffers required [65]. Ultrafiltration can be used for concentration, but direct use of filtered plant extract has been reported for VLP vaccines, which would provide a cost-effective method for vaccine preparation [66,67].

Vaccine candidates are finally assessed using model or target animal species to demonstrate their efficacy and safety [68]. The efficacy of vaccine candidates is thoroughly evaluated based on the ability to confer protection against viral challenge. The ideal veterinary viral vaccines should be able to induce humoral and T-cell-mediated immune responses that are able to prevent viraemia or reduce viral loads to prevent virus shedding. A safety profile of vaccine candidates must be evaluated at a very high dose (overdose) on the most susceptible animals by different routes of administration [69]. A safe vaccine is expected to exhibit low or zero adverse reactions.

### 5.1. Plant-Based Vaccines against Avian Influenza

Most published studies describing plant expression for AIV vaccine preparation have used on HPAI H5 HA as a prototypical antigen. This is likely due to the interest in developing such vaccines for human administration, with the potential for H5 to be a serious zoonotic disease. Thus, mice rather than chickens are often used as a model for assessing immunogenicity and protective efficacy of the vaccines. For poultry, vaccine development aims to protect against a range of HPAI and LPAI viruses including HPAI viruses H5 and H7, and LPAI H9N2 [4]. Examples of avian influenza plant-based vaccines are summarised in Table 3.

Early studies using transient expression in *N. benthamiana* focused on the expression of full-length HA. For example, the HA of an H7 AIV synthesised with modifications including a Kozak sequence, PR1a signal peptide, a C-terminal 6xHis tag for purification, and an endoplasmic retention signal (SEKDEL) was cloned into the pEAQ-HT vector, which was then co-infiltrated with a vector (pJL3:p19) containing the viral gene-silencing suppressor p19 from *Tomato bushy stunt virus* [70]. Gene expression was measured by RT-qPCR to determine that mRNA expression peaked at day 6 post-infiltration. On the other hand, in a study comparing expression in alfalfa, lettuce, and soybean, despite having the highest transcription rate determined by RT-qPCR, lettuce had the lowest protein expression levels [71]. The authors also compared different signal peptides to target expression to different cell compartments (the ER, apoplastic spaces or protein bodies) and found the highest level of expression in alfalfa leaves with a signal peptide targeting the apoplastic space. Mortimer et al. [72] also used vectors that targeted expression to different cell compartments (apoplastic spaces, chloroplasts and cytoplasm) and compared transient (in *N. benthamiana*) and transgenic (in *N. tabacum*) expression of full-length HA and HA missing the transmembrane domain. The study found that for the full-length HA, expression was highest in the apoplastic spaces, whereas the truncated form accumulated at higher levels in the ER. The authors speculated that the SEKDEL-tagged H5 may be less stable than the tagged truncated form. They were able to extract apoplast-targeted H5 protein by simply infiltrating the leaves with buffer containing Triton X-100 followed by low-speed centrifugation of whole leaves. Avoiding the need to homogenise the leaves reduces contamination with plant intracellular proteins and insoluble plant material. In another study, expression of the extracellular domain of an H5 HA was targeted to *N. tabacum* seeds using two regulatory sequences of seed storage protein genes from *Phaseolus vulgaris* [73]. The rationale for this was that due to their function, seeds accumulate high levels of protein and dehydration enables longer-term storage with reduced protein degradation. Chickens were immunised with two doses, 28 days apart, of crude seed extracts with Montanide adjuvant. Antibody responses, measured by haemagglutination inhibition, were predicted to be sufficient to be protective; however, no virus challenge was performed, and the responses were not compared with chickens immunised with leaf-produced protein.

The immunogenicity of recombinant proteins can be improved by oligomerisation, for example, by attaching proteins to nanoparticles using SpyTag/SpyCatcher technology [74]. Disadvantages of this technology are that it requires purified antigens and is patented. Lower-cost alternatives to generate oligomers have been explored. Fusion of the ectodomain of H5 HA with the intrinsic trigger motif of GCN4 enables the expression of HA trimers. The addition of the tailpiece sequence from the C-terminal sequence of mouse IgM (which cross-links trimers via disulphide bonds), homo-antiparallel peptides or homodimer proteins leads to oligomerisation of the trimers. Crude leaf extracts were used to immunise mice; extracts containing HA oligomers were found to induce higher titres of neutralising antibody than extracts containing HA trimers [75]. Similarly, more chickens survived a lethal challenge after immunisation with crude leaf extracts containing HA oligomers than with trimers [67].

**Table 3 vaccines-10-00478-t003:** Examples of avian influenza plant-based vaccines.

Antigen	Expression Host	Immunity	Notes	References
HA (H7)	*N. benthamiana*	Not mentioned	Transient expression	[70]
HA1	Alfalfa, soybean, and lettuce	Not mentioned	Expression targeted to ER, apoplastic space and protein bodies	[71]
Full-length and truncated HA (H5)	*N. benthamiana*, *N. tabacum*	Specific immune response in mice and chicken	Targeting of different cell compartments	[72]
HA (H5)	*N. tabacum*	HAI antibody raised in chickens	Targeted expression in seeds	[73]
Oligomeric HA (H5)	*N. benthamiana*	Neutralising antibody in chicken and mice	Oligomerisation achieved by several approaches	[75,76,77]
HA+M2 VLPs (H6)	*N. benthamiana* ^1^	Neutralising antibody in chicken	Co-expression of M2 and HA increased yield of VLPs	[78]
HA	*Arabidopsis thaliana*	HA-specific antibodies and mucosal antibodies in mice	Oral administration	[79]
Truncated HA (H5)	*N. benthamiana*	Specific antibodies in mice and ferrets	Intranasal administration	[80]
M2e peptide (H5)	*N. tabacum*	Not mentioned	Stable transfection	[81]
M2e fused to ricin toxin B chain (H5)	Duckweed	Specific antibodies against M2e peptide in mice	Oral administration to mice	[82]
HA (H5)	Duckweed	Neutralising antibodies and protective immunity in birds	Oil-in-water emulsion protected chickens	[83]

^1^*N. benthamiana* lacking plant-specific N-glycan residues.

Although HA will self-assemble to produce VLPs, it was reported that infiltration of equal amounts of *A. tumefaciens* transformed with the pEAQ-*HT* vector expressing HA and M2 enhanced VLP production compared to HA alone [78]. The authors also used *N. benthamania* plants engineered to significantly reduce xylosylated and/or core alpha1,3-fucosylated glycan structures resulting in more vertebrate-like N-glycosylation patterns [80]. A single dose of VLPs formulated with Montanide adjuvant elicited comparable immune responses, measured by haemagglutination inhibition, and better protection in chickens, against challenge with a heterologous virus, compared to two doses of a commercially available traditional inactivated oil-in-water vaccine. The authors estimated that 1 kg of plant material would produce sufficient doses of VLP vaccine for the immunisation of up to 30,000 chickens; such a cost-effective approach is vital to allow the widespread use of vaccines for poultry, particularly in resource-poor settings.

In most studies that tested the protective efficacy and/or immunogenicity of plant-expressed proteins in vivo, the protein is administered by intramuscular (IM) immunisation with an adjuvant. However, the induction of an immune response including secretory immunoglobulin A (sIgA) antibodies at mucosal surfaces, where pathogens such as AIV and NDV invade, is expected to afford better protection than a purely systemic immune response. One study demonstrated that oral administration of freeze-dried leaf powders from transgenic *A. thaliana* expressing H5 HA in the ER induced robust immune responses in mice and protected them from lethal challenge [79]. They found that saponin was more effective as an adjuvant than cholera toxin or flagellin. In another study, mice and ferrets inoculated intranasally with truncated H5 HA protein produced by transient expression of *N. benthamiana* with a mucosal adjuvant (c-di-GMP) were protected from virus challenge [80].

There have been numerous reports of the expression of M2e peptide in plants (e.g., [81]). The 23 amino acid peptide is the extracellular domain of M2 and is highly conserved among all influenza A viruses, making it a candidate for a universal vaccine (i.e., effective against all subtypes). Tarasenko et al. [81] found that expression failed if a longer sequence (43 amino acids) was encoded, presumably because this included the transmembrane domain, which forms an ion channel. However, short peptides are usually poorly immunogenic. Oral immunisation with four doses of the M2e peptide fused to ricin toxin B chain in nuclear-transformed duckweed induced a specific antibody response (measured by ELISA) in mice [82]. However, it was necessary to administer partially purified preparations by gastric gavage because the mice would not eat the duckweed, whether it was fresh or lyophilised, even when mixed with dry feed. On the other hand, supplementation of chicken feed with duckweed is advocated, particularly for developing countries, as a source of protein instead of soya bean or fishmeal supplements [84]. A further advantage of duckweed is that it can be cultivated in bioreactors, providing a relatively contained system that minimises the potential for accidental release of genetically modified (i.e., stably transfected) plants into the environment.

In recent years, there has been increasing awareness of the importance of antibodies to NA as a component of a protective immune response [85]. Although antibodies to NA do not prevent infection, they can significantly reduce disease severity and can provide heterologous immunity. Similarly, the potential value of including the highly conserved NP in vaccines to provide cross-protective T-cell-mediated immunity in the event of an antigenic mismatch between the HA of vaccine and circulating HA has long been recognised [86], though this has been explored less in relation to plant-based vaccines.

### 5.2. Plant-Based Vaccines against Newcastle Disease Virus

The production of NDV VLP vaccines has yet to be implemented in plants. Nevertheless, plant expression systems have been shown to be a robust system for the production of NDV structural proteins, particularly glycoproteins F and HN (Table 4). Plant expression systems can ensure proper folding and post-translational modifications of NDV glycoproteins. These are crucial factors for future development of plant-based NDV VLP.

Initially, recombinant HN protein was produced using transgenic plant cell cultures. In 2006, Dow AgroSciences received approval from the United States Department of Agriculture (USDA) for the first HN-based NDV vaccine produced in a tobacco-derived suspension cell line, although they did not bring the vaccine to market. Three doses of the vaccine injected subcutaneously was reported to show 90% protection against lethal NDV challenge in chickens [87].

**Table 4 vaccines-10-00478-t004:** Plant-based vaccines against Newcastle disease.

Antigen	Host	Notes	References
HN (LaSota)	*N. benthamiana* ^1^	Compared SP of HN with sea anemone equistatin	[88]
HN (LaSota)	*N. tabacum*	N/A	[89]
F (LaSota)	*Zea mays*	Neutralising antibodies elicited in chickens	[90,91]
F and HN (LaSota)	*Solanum tuberosum* cv. Kennebec	Humoral and cell-mediated immune responses in mice	[88,92]
F and HN (LaSota)	*N. tabacum*	ELISA ^2^ detection of anti-NDV antibodies in Rabbit	[93,94]
F and HN (chicken/SPVC/Karachi)	*Zea mays*	Specific locally secreted IgY in chickens	[95]
F (XJ-2/97)	*Oryza sativa*	Challenge of chickens	[96]

^1^ Transient expression. ^2^ ELISA: enzyme-linked immunosorbent assay.

Attempts have also been made to express the NDV HN protein using *Agrobacterium*-mediated transient expression systems. Gomez et al. (2009) tested several binary vector constructs that varied with regards to a substitution of the native N-terminal signal peptide (SP) of HN with the signal peptide of sea anemone equistatin (SPE) and addition of KDEL at the C-terminus [88]. The construct with a native signal peptide and KDEL sequence resulted in the highest protein yield. Interestingly, the substitution of a native signal peptide with SPE was reported to disrupt protein folding and resulted in a low yield of recombinant HN. The lack of plant-derived glycans, a result of inclusion of the KDEL sequence, was indicated by a qualitative assay using PNGase F, an endoglycosidase that cleaves oligosaccharides from N-linked glycoproteins [81].

The application of transgenic plants as edible vaccines has also been explored for ND prevention. However, immune tolerance has long been considered a potential barrier for development of a protective immune response from edible vaccines. Immune tolerance may be an over-estimated problem for some species due to the ease of inducing oral tolerance in the most common laboratory animals such as mice and rats. Interestingly, oral tolerance has been shown to be much less of an issue in poultry older than 3 days of age, potentially making edible vaccines more suitable for use in poultry [97]. Hahn et al. [89] explored the use of transgenic tobacco for developing an edible vaccine for ND. In their system, the DNA construct was derived from plant binary vector p221 and the expression of the HN gene was driven by *Cauliflower mosaic virus* (CaMV) 35S promoter. Leaf disc transformation of tobacco was carried out using *A. tumefaciens* EHA105 strain. Unfortunately, immunisation of 6-week-old chickens using lyophilised tobacco leaf extract failed to induce production of neutralising antibodies [89].

In contrast, animal studies suggested that recombinant HN and F expressed in transgenic potato were able to induce a mucosal immune response [92,98]. Adult Balb/c mice were fed with potato leaves expressing either recombinant HN or F protein. Intestinal washes were collected on day 36 and further analysed using ELISA and immunofluorescence assays. The results indicated the presence of NDV-specific IgA secreting plasma cells in the intestinal tissue that was comparable to the response induced by oral immunisation with LaSota virus-soaked potato leaves. Furthermore, oral immunisation of mice with transgenic potato expressing HN and F protein was also able to induce the production of locally secreted IgY (avian IgG equivalent) and CD8+ T cells [92]. Similarly, the production of locally secreted IgY was observed in chickens orally fed with transgenic corn expressing F and HN protein [95].

The recombinant F protein has been successfully expressed and extracted from other staple crops such as transgenic rice and corn [90,91,96]. The extracted F protein was able to induce the production of neutralising antibodies post-challenge with a velogenic strain in chickens [90,91]. The survival rates of chickens immunised with F protein were comparable to those immunised with LaSota virus vaccine.

Another approach suggested the feasibility of producing concatenated F and HN epitopes without a genetic fusion to M protein in transgenic tobacco [94]. The authors established a DNA construct with four 96 base-pair tandem repeats of an HN epitope followed by three 153 base-pair tandem repeats of an F epitope. An N-terminal 6xHis tag and omega sequence as a ribosome binding site were attached upstream of the DNA construct. Lastly, the authors also included the ER retention signal SEKDEL. In follow-up studies, the authors reported that the transgenic plant extract was only moderately immunogenic in rabbits. Using ELISA, the transgenic plant extract was shown to induce lower production of anti-NDV antibodies in comparison to commercial NDV vaccines [94].

Some, but not all, viral glycoproteins are expressed at a low yield in plant transient systems [99]. The influenza glycoprotein seems to be an exception to this issue. In contrast, the expression of NDV HN from a velogenic strain using pEAQ-HT expression vector shows a relatively low yield with plants showing severe tissue necrosis starting at day 5 post-infiltration (Nurzijah, unpublished data). The combination of low expression and tissue necrosis in transient plant expression systems may be related to ER stress [100,101]. Complex viral glycoproteins undergo post-translational modification in the ER; discrepancies between viral native host and plant cellular machinery may contribute, in some instances, to improper N-glycosylation of viral glycoproteins [102,103,104]. N-glycans are recognised as a signal to the ER quality control machinery in eukaryotic organisms. ER quality control comprises enzymatic constituents: glycoprotein glucosyl transferase (UGGT) and glucosidase II as well as ER chaperone proteins. This cellular machinery will retain misfolded secretory proteins in the ER [105,106,107], and the accumulated misfolded protein will undergo degradation, and hence, low protein expression in plants. In theory, the amount of misfolded protein can be reduced by extending the transit period of newly transcribed protein in the ER. This has been carried out by targeting viral glycoproteins to sub cellular organelles with the conjugation of C-terminus ER retention signal (KDEL/SEKDEL) or replacing protein native signal peptide using those originated from plants [85]. Recently, the use of heterologous chaperone proteins is receiving attention. Co-expression of human chaperone proteins calnexin and/or calreticulin was able to improve the expression of human immunodeficiency virus (HIV) type 1 soluble gp140 in a *N. benthamiana* transient expression system. The benefit of this approach was also observed for other viral glycoproteins derived from Epstein–Barr virus, Rift Valley fever virus and Chikungunya virus [103].

## 6. Advantages of Plant-Based Vaccines

Ultimately, the future success of plant-based vaccines depends on their ability to protect against infectious disease in a safe and cost-effective manner, and offer advantages compared to conventional vaccine platforms. A number of studies have shown the protective ability of plant-expressed vaccines to be equal to or greater than conventional vaccines. In one study, H5 (HA) was synthesised from A/chicken/Indonesia/7/2003 (H5N1) (Indo/03) and expressed in duckweed. Birds immunised (IM) with the recombinant H5 protein were fully protected against the homologous HPAI H5N1 challenge, with partial protection against heterologous virus challenge. The protection afforded by the recombinant proteins was similar to a commercial inactivated whole virus vaccine [83]. In another trial, H5 HA (H5N1) oligomers and trimers were transiently expressed in *N. benthamiana*, and the proteins were extracted and purified. Chickens were immunised with the purified oligomers and trimers and survived against lethal virus challenge with a protection rate of 92% for oligomers and 75% for trimers [75]. Regarding VLPs, H6 HA was co-expressed with M2 in *N. benthamiana*. Birds were vaccinated twice (IM), with the partially purified VLPs, which induced a high titre of antibody measured by HI. Virus shedding was reduced significantly following heterologous LPAI H6N2 challenge in comparison with unvaccinated birds or those vaccinated with a conventional whole inactivated H6N2 oil-emulsion vaccine prepared using eggs [78].

The production of traditional vaccines can be very expensive, especially the production of killed virus vaccines for HPAI because of the requirement for high levels of containment due to the zoonotic risk. In contrast, plant vaccine production has relatively low costs. The plants are cheap, able to be grown relatively easily, and offer a safe production system for recombinant proteins. The design of target gene(s) is now relatively trivial using gene synthesis, which is now more affordable than before. The production of vaccine constructs using gene synthesis allows the incorporation of features to improve the protein expression as outlined above. The infiltration of plants is an easy method, with no need for expensive equipment, allowing rapid vaccine production. The administration of plant vaccines can be performed either parenterally or orally. Oral administration in feed allows vaccines to be administered with relatively little purification, thus reducing the associated costs [108,109]. Edible vaccines, administered orally, can generate both systemic and mucosal immunity. Plant vaccines can be stored at room temperature [110]. Yields of recombinant proteins, when optimised, can be high [78]. Plants ease mass production without the need for complicated procedures and equipment, which will be helpful for use in developing countries [110].

The ability to produce vaccines that do not require cold-chain storage and transportation is highly desirable. There has been some success in realising this for plant-based vaccines, for example, a tuberculosis vaccine expressed in tobacco or lettuce was stable for six months at room temperature following lyophilisation [111]. Plant vaccines are free from toxic substances and animal pathogens [112]. Plants have their own machinery for assembly, folding, and glycosylation of the recombinant protein; generally, these do not have the disadvantages associated with the glycosylation observed in yeast or bacterial systems [113]. Furthermore, there have been significant efforts to engineer plants that show a more mammalian glycosylation pattern [114].

## 7. Challenges for Oral Delivery of Plant-Based Vaccines?

Determining and delivering an appropriate dose of vaccine may be difficult, especially when vaccines are administered orally. The effective dose may differ according to the type of the vaccine, age, and weight of the bird. Multiple doses with the vaccine may lead to immune tolerance, and a lower dose leads to poor immunogenicity. However, delivery of vaccines via injection is not ideal in the majority of poultry-rearing systems due to the high number of birds, but it may be suitable for small numbers of birds in backyard poultry flocks [115].

Oral (immune) tolerance has been observed following exposure to or feeding of protein antigens to animals. When an antigen is presented among the intestinal contents to the mucosal immune system, it may induce oral tolerance, likely due to the activity of T regulatory cells [116]. This may subsequently lead to a suboptimal immune response if the body encounters the same antigen again during natural infection. Mucosal immune tolerance has therefore been highlighted as a significant hurdle that must be overcome by oral vaccines. [117]. Oral adjuvants have been used to try to overcome this problem, but this raises concerns over the potential development of allergies to other food proteins [118]. VLPs do not need any adjuvant and may therefore overcome this problem, perhaps because VLPs are able to directly stimulate dendritic cells, leading to antigenic recognition and the activation of T and B cells [27]. It is worth emphasising that most studies on oral tolerance have been performed on mice or other mammals. Studies performed on chickens have shown that oral tolerance can be induced in 1–3-day-old chicks, but in older chicks, recombinant proteins elicit solid immune responses rather than tolerance [97].

## 8. Conclusions and Future Prospects

The prevalence of AI and ND infections in poultry worldwide poses a great risk for food security, particularly in low- and middle-income countries. Rapid and cost-effective vaccine production is certainly a pre-requisite for managing future outbreaks of poultry viral diseases such as AI and ND. Recombinant AIV and NDV proteins expressed in plants can induce humoral and T-cell-mediated immune responses comparable to commercial vaccines. Additionally, transient plant expression systems can properly assemble influenza HA-based VLPs, which further emphasise its role as a suitable host for production of VLP-based vaccines [119,120]. Although stable or transient gene expression can be used, transient expression systems are favoured for rapid protein production. The application of disarmed *A. tumefaciens* for plant transformation has ensured high yields of recombinant viral proteins [45]. Plant vaccines are safe and the post-translational machinery ensures the immunogenic properties of the recombinant protein are authentic [113]. Challenges related to expression of viral glycoproteins have recently begun to be addressed using chaperone proteins. Some plant vaccines can be lyophilised, allowing them to be stored and distributed at room temperature [110]. The prospects for oral delivery of plant-derived vaccines in poultry is enticing, offering the potential to reduce greatly the costs of purification [97,108,109]. The use of deconstructed expression systems derived from plant viruses, which can carry larger genes, has circumvented many problems of low protein production [50]. While the development of cheap and effective plant-based vaccines faces developmental challenges, there have been notable successes, which should encourage optimism for their future prospects.

## Figures and Tables

**Figure 1 vaccines-10-00478-f001:**
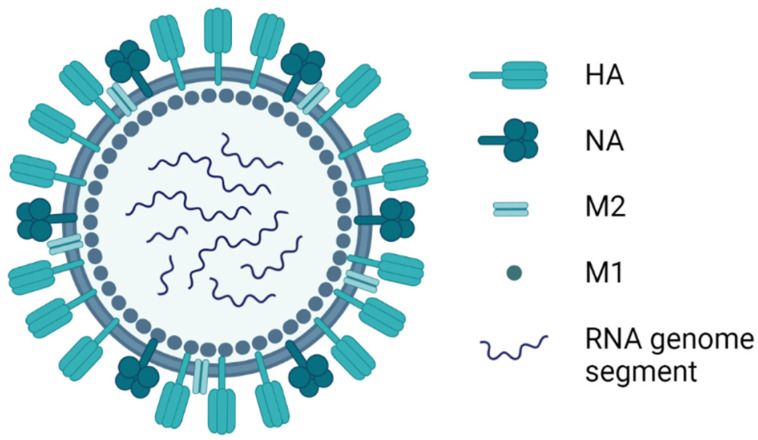
Graphical representation of the avian *Influenza A virus* particle. HA, haemagglutinin, NA, neuraminidase, M2, matrix protein 2, M1, matrix protein 1. Image created using BioRender.com.

**Figure 2 vaccines-10-00478-f002:**
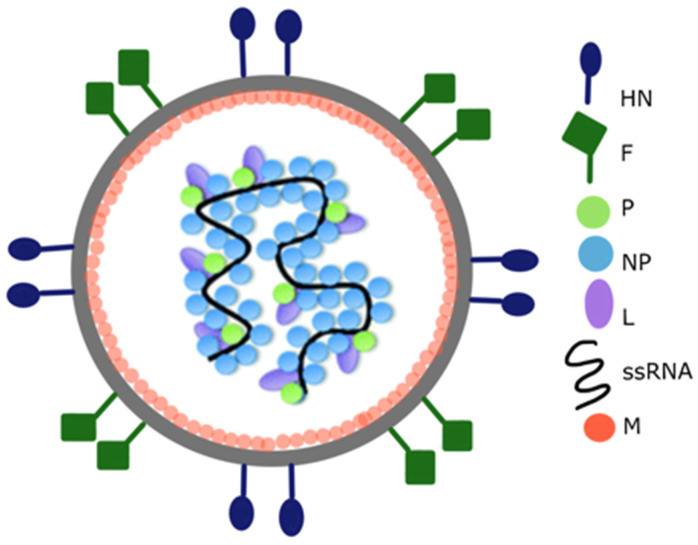
Graphical representation of the Newcastle disease virus particle structure. Anchored to the surface of the virus particle envelope are haemagglutinin-neuraminidase (HN) and fusion (F) glycoproteins. Matrix (M) proteins are peripherally attached to the NDV envelope. The interior of the virus particle is composed of negative-sense single-stranded RNA and RNA-associated nucleoprotein (NP), phosphoprotein (P), and large polymerase (L). Adapted from [9].

**Table 2 vaccines-10-00478-t002:** List of commonly used RNA and DNA plant virus-based deconstructed vectors for *Agrobacterium tumefaciens*-mediated transient transformation *in planta*.

Vector Name	Vector Backbone	Features and Modification(s)	References
Magnifection system developed by Icon Genetics, Germany (MagnICON)	Hybrid between TMV and TVCV	Consist of 3 modules: 5′ module: *Arabidopsis* actin 2 (ACT2) promoter, TMV polymerase and movement protein genes; 3′ module: gene of interest (GOI) fused to viral coat protein (CP) and nos terminator; PhiC31 integrase gene (from *Streptomyces* phage C31)	[50,53]
pEAQ-HT	CPMV	Based on a deleted version of CPMV RNA-2 and uses the 5′ and 3′ UTRs from CPMV RNA-2 Removal of the upstream AUG codons of 5′ UTR to enhance translational efficiency Incorporation of p19 sequence from Tomato bushy stunt virus into the T-DNA region	[41,52,54]
pHREAC	CPMV	Modification of synthetic 5′ UTR from CPMV-HT	[43]
TRBO (TMV-RNA-overexpression)	TMV	Deletion of TMV splicing sites Addition of RNA-dependent RNA-polymerase (RdRp) Deletion of TMV CP to allow addition of GOI(s) beside the 3′ UTR	[55]
pBID4	TMV	Consist of binary vector pBI121 and TMV Replacement of TMV CP gene by the GOI(s) resulting in self-replicating vector	[56]
PVX vector	PVX	Removal of PVX CP gene and triple gene block Addition of Potato virus A suppressor of gene silencing Addition 5′ UTR upstream GOI open reading frame	[57]
BeYDV	Geminivirus	Deletion of BeYDV movement and CP genes Addition of p19 gene into the expression cassettes	[58]

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
