# Peer review of "Development of Plant-Based Vaccines for Prevention of Avian Influenza and Newcastle Disease in Poultry"

_vaccines, 2022, doi:10.3390/vaccines10030478_

Round 1
Reviewer 1 Report
There is no problem in this review itself, and the content of the review is relatively detailed, but from the perspective of academic thought,We must comprehensively describe the possible problems of the vaccine from various angles.
First: plant vaccines still belong to VLP vaccines and subunit vaccines. The defects of VLP vaccines and subunit vaccines may not be described in detail. We must admit that we do not know all the detailed biological functions of viral proteins (structural proteins and non structural proteins), but only know some functions, which is a fact. There are many studies on the function of nonstructural proteins in virus escaping host immune response, but there are few studies on the function of structural proteins in virus escaping host immune response, but the immunosuppressive function of structural proteins does exist or has been ignored by vaccine developers. I believe that there are many unpublished or unconfirmed problems in VLP vaccine and subunit vaccine.I believe that VLP vaccine and subunit vaccine are by no means the development direction of vaccines in the future, including plant vaccines. Second: plant vaccine also has many problems as an oral vaccine, The immune tolerance of animals to food is obviously impossible for extensive and long-term application. So, the suggestion by some authors that “the future is plants” may be an understatement.
Author Response
Reviewer comment: “There is no problem in this review itself, and the content of the review is relatively detailed, but from the perspective of academic thought, we must comprehensively describe the possible problems of the vaccine from various angles”
Author response: Thank you. We agree with the reviewer’s assessment that we may have neglected to comprehensive evaluate the potential problems associated with some of the vaccine types described in this manuscript. We have revised the manuscript accordingly. The main changes are detailed below and highlighted with “track changes” in the text.
Comment 1 from reviewer noting that:
a. The defects of VLP vaccines and subunit vaccines may not be described in detail.
b. The development of plant vaccines derived from viral structural proteins (VLP and subunit vaccines) may not be immunologically relevant considering the detailed biological functions of viral proteins remains elusive.
c. VLP and subunit vaccines are by no means the development direction of vaccines in the future, including plant vaccines.
Author response:
a. Thank you for pointing this out. Regarding the VLP vaccines we have added information related to stability issue for balancing the benefits described in the manuscript.
“The technical challenges for generation of VLPs from enveloped viruses (enveloped VLPs/eVLPs) such as AIV and NDV, remain a major hurdle to the success of this technology going forward. The stability of eVLPs requires thorough considerations of the appropriate temperature, shear force, and processes that are used in purification. Improper purification may lead to degradation of VLPs and a consequent reduction in immunogenicity. Most often, this leads to more complex downstream processing than conventional vaccines [30]. Alternative approaches for the assembly of VLPs are also being investigated, including the development of “hybrid” VLPs which use the core of one virus as a scaffold for the surface glycoproteins of another virus. Hepatis B cores (HBc) are one example of this, where the HBc structure has been used as a VLP core to present surface antigens from heterologous viruses including HIV, Foot and mouth disease virus, and influenza A (reviewed in [31]). Such hybrid VLPs may offer a universal platform for VLP production and have the advantage of producing non-enveloped particles which are inherently more stable than enveloped VLPs [32].” (Page 7, section: Overview of vaccines for AI and ND immunisation)
The following information is also added regarding subunit vaccines:
“However, subunit vaccines have a more restricted antigenic repertoire than attenuated viral vaccines; as such it is important to ensure that the elicited immune responses provide robust protection against viral challenge. For example, an ND subunit vaccine that is derived from a single glycoprotein may be poorly immunogenic; the use of both NDV F and HN leads to a broader antibody response that provides greater protection against viral challenge [21].” (Page 5, section: Overview of vaccines for AI and ND immunisation).
b. Thank you for the reviewer’s assessment.
For AI, we added an immunity section in the AI examples’ table (Page 12, Table 4. Examples of avian influenza plant-based vaccines).
We also highlighted the results of the challenge experiment with plant-based vaccines in comparison with the inactivated vaccines. The additional information is as follows:
“Ultimately, the future success of plant-based vaccines depends on their ability to protect against infectious disease in a safe and cost-effective manner, and offer advantages compared to conventional vaccine platforms. A number of studies have shown the protective ability of plant expressed vaccines to be equal to or greater than conventional vaccines. In one study, H5 (HA) was synthesized from A/chicken/Indonesia/7/2003 (H5N1) (Indo/03) and expressed in duckweed. Birds immunised (IM) with the recombinant H5 protein, were fully protected against the homologous HPAI H5N1 challenge, with partial protection against heterologous virus challenge. The protection afforded by the recombinant proteins was similar to a commercial inactivated whole virus vaccine [83]. In another trial, H5 HA (H5N1) oligomers and trimers were transiently expressed in N. benthamiana and the proteins extracted and purified. Chickens were immunized with the purified oligomers and trimers and survived against lethal virus challenge with a protection rate of 92% for oligomers and 75% for trimers [75]. Regarding VLPs, H6 HA was co-expressed with M2 in N. benthamiana. Birds were vaccinated twice (IM), with the partially purified VLPs, which induced a high titre of antibody measured by HI. Virus shedding was reduced significantly following heterologous LPAI H6N2 challenge in comparison with unvaccinated birds or those vaccinated with a conventional whole inactivated H6N2 oil-emulsion vaccine prepared using eggs [78].” (Page 16, section: The advantages of plant-based vaccines).
c. We agree with reviewer’s argument regarding subunit vaccines. Plant-based subunit vaccines are by no means is superior to commercial AI and ND vaccines in all respects. Thereby throughout the manuscript we re-clarify the focus of our review, which is incorporating plant expression system(s) for the development of VLP vaccines against AI and ND.
The changes that have been made are as follow:
“In particular, we highlight the development of plant-based virus-like particle (VLP) vaccines against AI and ND.” (Page 2, section: Introduction)
“The production of NDV VLP vaccines has yet to be implemented in plants. Nevertheless, plant expression systems have been shown to be a robust system for the production of NDV structural proteins, particularly glycoproteins F and HN (Table 4). Plant expression systems can ensure proper folding and post-translational modifications of NDV glycoproteins. These are crucial factors for future development of plant-based NDV VLP.” (Page 13, sub-section: Plant-based vaccines against Newcastle disease virus)
Plant based VLP vaccines have shown significant promise and are under development by a number of large pharma worldwide. We have highlighted this relevant example:
“However, the technology for eVLPs production has shown significant advances. Recently, the Canadian biotechnology company Medicago reported that Health Canada approved the new drug submission (NDS) for a plant-based seasonal human influenza recombinant vaccine for scientific review. This is a quadrivalent virus-like particle (QVLP) vaccine including the HA of two influenza A virus subtypes and two lineages of influenza B virus transiently expressed using Agrobacterium tumefaciens-mediated transient transformation (also known as ‘agroinfiltration’). The QVLP vaccine elicited both homologous and heterologous antibody responses at higher doses, with the introduction of adjuvant having little or no effect. In the first phase III efficacy study, the candidate vaccine also reliably elicited both homologous and heterologous CD4+ T helper cells identified by generation of interferon-gamma (IFN-γ), interleukin-2 (IL-2) and/or tumour necrosis factor alpha (TNF-α) [33].”(Page 7, section: Overview of vaccines for AI and ND immunisation).
Reviewer Comment: This should be evidence for future prospect of plants-based VLPs for prevention of AI and NDV.
We have added a section to highlight potential advantages of plant-based vaccines. The additional information is as follows:
“The production of traditional vaccines can be very expensive, especially the production of killed virus vaccines for HPAI because of the requirement for high levels of containment due to the zoonotic risk. In contrast, plant vaccines production has relatively low costs. The plants are cheap, able to be grown relatively easily, and offer a safe production system for recombinant proteins. Design of target gene(s) is now relatively trivial using gene synthesis, which is now more affordable than before. Production of vaccine constructs using gene synthesis allows the incorporation of features to improve protein expression as outlined above. Infiltration of plants is an easy method, with no need for expensive equipment, allowing rapid vaccine production. Administration of plant vaccines can be performed either parenterally or orally. Oral administration in feed, allows vaccines to be administered with relatively little purification, thus reducing associated costs [108,109]. Edible vaccines, administered orally, can generate both systemic and mucosal immunity. Plant vaccines can be stored at room temperature [110]. Yields of recombinant proteins, when optimized can be high [78]. Plants ease mass production without the need for complicated procedures and equipment which will be helpful for use in developing countries [110].
The ability to produce vaccines that do not require cold-chain storage and transportation is highly desirable. There has been some success in realizing this for plant based vaccines, for example a tuberculosis vaccine expressed in tobacco or lettuce was stable for six months at room temperature following lyophilisation [111]. Plant vaccines are free from toxic substances and animal pathogens [112]. Plants have their own machinery for assembly, folding, and glycosylation of the recombinant protein; generally these do not have the disadvantages associated with the glycosylation observed in yeast or bacterial systems [113]. Furthermore, there have been significant efforts to engineer plants that show a more mammalian glycosylation pattern [114]. (Page 16, section: The advantages of plant-based vaccines).
2. Reviewer comment: noting the issues related to immune tolerance of oral or edible vaccine making it impossible for extensive and long-term application.
Author response: The majority of concerns around oral immune tolerance associated with oral delivery of vaccines has arisen from work in mammals. In chickens, there is evidence that this may be less of a concern, provided animals are immunised at older than 3 days. Nevertheless, we see the value in discussing the potential for oral tolerance and have added the following text:
“Determining and delivering an appropriate dose of vaccine may be difficult, especially when vaccines are administered orally. The effective dose may differ according to the type of the vaccine, age, and weight of the bird. Multiple doses with the vaccine may lead to immune tolerance, and a lower dose leads to poor immunogenicity. However, delivery of vaccines via injection is not ideal in the majority of poultry rearing systems due to the high number of birds, however may be suitable for small numbers of birds in backyard poultry flocks [115].
Oral (immune) tolerance has been observed following exposure or feeding of protein antigens to animals. When an antigen is presented among the intestinal contents to the mucosal immune system, it may induce oral tolerance, likely due to the activity of T regulatory cells [116]. This may subsequently lead to a suboptimal immune response if the body encounters the same antigen again during natural infection. Mucosal immune tolerance has therefore been highlighted as a significant hurdle that must be overcome by oral vaccines. [117]. Oral adjuvants have been used try to overcome this problem, but this raises concerns over the potential development of allergies to other food proteins [118]. VLPs do not need any adjuvant and may therefore overcome this problem, perhaps because VLPs are able to directly stimulate dendritic cells leading to antigenic recognition and activation T and B cells [27]. It is worth emphasising that most studies on oral tolerance have been performed in mice or other mammals. Studies performed in chickens have shown that oral tolerance can be induced in 1- 3 day old chicks, but in older chicks, recombinant proteins elicit solid immune responses rather than tolerance [97].”(Page 16, New section: Challenges for oral delivery of plant-based vaccines?).
Reviewer comment: I believe that there are many unpublished or unconfirmed problems in VLP vaccine and subunit vaccine. I believe that VLP vaccine and subunit vaccine are by no means the development direction of vaccines in the future, including plant vaccines.
Author response: Like all areas of science, unfortunately negative results often remain unpublished. There are likely to be negative results for all vaccine types. The authors did not intend to give the impression that VLP vaccines (or especially subunit vaccines) are likely to be successful against all viral diseases or superior to conventional vaccines in all scenarios. Development of vaccines will continue to require an individualised approach to each virus. However, there have been some significant early successes which provide optimism at least for influenza plant-based VLP vaccines (and to some extent for NDV). We hope that the amendments within the manuscript provide a more balanced viewpoint.
Reviewer comment: So, the suggestion by some authors that “the future is plants” may be an understatement.
This statement was a quotation from Peyret and Lomonossoff who have extensive experience with development of plant-based expression systems. Nevertheless we have amended this to: “While the development of cheap and effective plant-based vaccines faces developmental challenges, the suggestion by these authors that “the future is plants” encourages optimism for their future prospects.” (lines 639-641).
Reviewer 2 Report
Comments on manuscript:
I have noted major revision to catch your attention
- Although the review presents relatively promising information that would benefit further investigations. The method is not well written and needs improvement. However, I will suggest you provide a PRISMA diagram of your search strategy, results from each database you used, then elimination of duplicates then how you selected articles for inclusion/exclusion. Please provide sufficient methods how to conduct this review. Readers should be able to replicate your search to update it.
- Work on splitting your sentences, most of them are quite long and hence, lack cohesion. Overall, the manuscript needs significant and extensive revision.
- The organization of the manuscript needs significant improvement, and the presentation also needs major improvements as well.
- Please provide separate detailed table, so the reader can judge the quality of the evidence on the studies for which you derive the data of plant-based vaccines. This will the key contribution you make.
Author Response
1. Reviewer 2 suggests taking a systematic review approach by providing a PRISMA diagram and a table of quality of evidence.
Author response: Thank you for this suggestion. It would certainly have been interesting to explore the effectiveness of different types of vaccine in poultry. However, this was never our intention. Indeed, we are aware of the inherent difficulties in making comparisons between different vaccination studies due to the lack of consistent methods employed. Studies described in the literature show frequent differences that make it extremely difficult to make comparisons between them. For example, there are differences in route of vaccination (oral, parenteral, intraocular etc), in dose of vaccine and choice of adjuvant, in the number of doses given, the choice and dose of challenge virus (if challenge is actually performed) and measures to assess protection (e.g. PCR methods versus virus isolation). Breed, age and source of animals may lead to significant differences when using outbred animals. There are also inherent caveats when trying to compare data from mouse models to that obtained in the natural (chicken) host. Therefore, we decided at the outset that a systematic review, employing recommended methods (pre-defined search criteria, exclusion criteria etc) would not be appropriate.
2. Reviewer 2 suggests splitting the long sentences for improving the cohesion.
Author response: Thank you for this suggestion. We have worked through the manuscript to revise longer sentences to improve the readability and coherence of the manuscript.
Reviewer 3 Report
The review paper is focused on the development of plant-based vaccines for prevention of avian influenza and Newcastle disease in poultry. The topic of the review is interesting considering the high cause of morbidity and mortality in poultry, thus resulting in significant economic losses. Notably, there are multiple factors that negatively affect the current administrations of vaccines e.g. prohibitive costs or poor protection against local emerging strains. In this context, the development of plant-derived vaccines may offer good stability and safety.
Overall, I consider the review quite interesting and complete. I have only a few minor remarks.
- In the Introduction it should be reported that besides viral diseases, also bacterial ones can affect poultry. For instance, psittacosis, also called ornithosis or parrot disease, is an infection that humans usually get from birds, due to the bacterium Chlamydia psittaci.
-I suggest the authors not to use sub-sections in the Introduction; as a consequence, they should enlarge it and devote a different section to avian influenza and Newcastle disease. All subsequent sections need to be re-numbered.
Author Response
“The review paper is focused on the development of plant-based vaccines for prevention of avian influenza and Newcastle disease in poultry. The topic of the review is interesting considering the high cause of morbidity and mortality in poultry, thus resulting in significant economic losses. Notably, there are multiple factors that negatively affect the current administrations of vaccines e.g. prohibitive costs or poor protection against local emerging strains. In this context, the development of plant-derived vaccines may offer good stability and safety.
Overall, I consider the review quite interesting and complete. I have only a few minor remarks”.
Author response: Thank you for your positive comments.
1. Reviewer 3 suggests for including bacterial infections in poultry, for instance, psittacosis, also called ornithosis or parrot disease, an infection that humans usually get from birds, due to the bacterium Chlamydia psittaci
Author response: We do think that it would be interesting to review the use of vaccines to control important bacterial diseases of poultry, which would include many pathogens for example also E. coli and Salmonella. However, this is outwith our area of expertise as virologists. It is also beyond the scope that we had agreed for our review: to cover the use of vaccines to prevent viral infection in poultry – specifically NDV and AI, in which we have active research projects.
2. Reviewer 3 suggests not to use sub-sections in the Introduction; as a consequence, the sections should be enlarged and devoted to a different section to avian influenza and Newcastle disease. All subsequent sections need to be re-numbered
Author response: Thank you for pointing this out. These corrections have been made in accordance with reviewer’s suggestion.
Round 2
Reviewer 1 Report
Very good, the authors have discussed and evaluated plant vaccines objectively and fairly.
Reviewer 2 Report
Dear Author
I recommend accepting your revised manuscript, all my comments have been addressed by authors.